# How Corruption Is and Should Be Investigated by Economic Theory

Petr Wawrosz 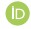

Faculty of Economics and Management, Czech University of Life Science, Kamýcká 129, 165 21 Prague, Czech Republic; wawrosz@pef.czu.cz

**Abstract:** The article analyzes how economic theory usually investigates corruption. It describes the main traditional economic theories dealing with this issue—principal-agent theory (agency theory) and rent-seeking theory—and it emphasizes that both face some problems, especially their neglect of some important factors as to why corruption occurs which prevents them from accurately analyzing this phenomenon and proposing solutions on how to fight against it. The article further discusses whether institutional economics can overcome these problems. We show that it does, but that it needs to more seriously consider the environment in which corruption occurs. Redistribution system theory can serve as a useful aid here because it reveals that the source of corruption is an environment of undesirable redistribution. The article provides the characteristics of this type of redistribution and shows how its reduction also leads to the reduction in corruption. It can be concluded that economic theory should not rely only on traditional approaches to corruption but should at least add institutional economics and redistribution system theory to its methods of corruption analysis.

**Keywords:** corruption; agency theory; rent-seeking theory; institutional economics; redistribution system theory

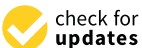



## 1. Introduction

The fight against corruption has received special attention in the last 20 or 30 years in comparison with the previous period (Prasad et al. 2019). Corruption and the questions of its prevention and reduction are now discussed from more than just the theoretical point of view. Many subjects, including the OECD, the World Bank, the United Nations, the European Union, and others, have adopted anti-corruption programs—their assistance to governments is for instance often conditional on the government's commitment to take concrete anti-corruption measures. What also changed was the approach to corruption. Previously, the focus was mainly on public sector corruption, and corruption was usually defined as "the misuse of public office for private gain" (e.g., World Bank 1997, p. 6). Today, it is recognized that corruption can affect all sectors and areas of life. Transparency International, the best-known non-governmental organization fighting against corruption, now defines it (Transparency International 2021a) as "the abuse of entrusted power for private gain". (For instance, Bowra et al. (2022); Dobrowolski et al. (2022); Stathopoulou et al. (2002) or Gisladottir et al. (2022) are among the recent authors using this definition). The word corruption became a universal term for many behaviors including bribery, clientelism, nepotism, illicit gifts, favors, patronage, informal and non-reported contacts, and so on. Corruption is seen as the main reason why many policies trying to improve the standard of living, including environmental issues, do not achieve their goals (Pozsgai-Alvarez 2020).

Breit et al. (2015, p. 319) and Torsello (2013, p. 313) mention that corruption literature is experiencing a boom, and corruption is examined from many different perspectives. The research not only tries to explain the reason for corruption, but it also theoretically and practically suggests and explains anti-corruption policies and ways by which corruption

could be reduced. Most of the literature concentrates on specific issues connected with the topic and does not discuss methodological questions including the approach of the relevant science to corruption. However, as Breit et al. (2015, p. 320) discuss, "what tends to be neglected is an investigation into, and thus understanding of, the underlying causes and mechanisms of the phenomenon". Each science must reflect and discuss how it investigates a specific topic. The prerequisites of the research must be critically evaluated and their possible weaknesses revealed, including the issue of how they affect research results. Such reflection and discussion can help us discover new ways to research and new issues for investigation. It may further contribute to answers to problems that have not been yet solved. The importance of methodological research is, in the case of corruption, emphasized by the fact that, as Prasad et al. (2019) warn, corruption is resistant, and it seems impossible to be tamed. If a theory is not able to precisely analyze corruption, it will not be able to offer successful means as to how to reduce it.

Although corruption is broadly discussed, there is no unique definition of corruption. It is probably impossible to achieve such a definition. Corruption is a too complex a phenomenon and the world is too heterogeneous a place for a generally accepted definition. We agree with Martinsson (2021, p. 269) that "To find a universal all-encompassing corruption definition would for corruption researchers be like finding the Holy Grail: it has not yet been done, and most likely never will be." However, without some of its characteristics, research is impossible, although a specific definition or understanding of the term frames and limits the scope of research. Different views can lead to different approaches to dealing with or fighting against corruption. The article recognizes that corruption involves a lot of phenomena that can be specified by other words (e.g., bribery, nepotism, etc.). We use the definition of Wawrosz (2019, pp. 270–71) and we understand corruption as "any behavior that deviates from generally accepted formal and informal obligations to achieve private gain. Corruption is a form of business. On the one hand, there is a subject possessing a certain form of power whether it is of action or decision. This person is called the corrupt agent and they can include more than one person (e.g., two or more government officials or two or more managers of a private firm). The corrupt agent is violating their duties and obligations because they or another subject (e.g., a corrupt relative) has or may benefit from a violation. On the other hand, there is a second subject called the corrupting agent (again several people, e.g., players of a sports club) can represent it. The corrupting agent provides or promises to benefit the corrupt agent if the corrupt agent violates their duty. The violation is beneficial for the corrupting agent. By the actions of the corrupt agent, the corrupting agent or some related agent such as a relative may gain some benefits at the expense of a third party that does not participate in corruption." To distinguish corruption from other forms of undesirable (immoral) behavior (such as fraud, which can include only the fraudster and the victim harmed by the fraudster), from our point of the view, the following features of corruption behavior should be emphasized: 1. Corruption always involves two parties with different roles (a corrupt agent and a corrupting agent); 2. Corruption harms a third party that does not participate in the corrupt behavior—because of the behavior, this party has a lower benefit or income, cannot achieve some opportunities, must incur greater costs, etc.; and 3. Corruption is a behavior contrary to universal principles. If everybody is corruptible, it makes human collaboration impossible.

The article aims to show how standard theories, specifically agency theory and rent-seeking theory research corruption, to reveal problems connected with the theories and to show how alternative approaches, mainly institutional economics and redistribution system theory, can overcome problems connected with standard theories. Its main contribution lies in the complex evaluation of standard and alternative approaches to corruption (to our best knowledge there is no other similar contemporary comprehensive article), in emphasizing the importance of institutional economics and redistribution system theory in the fight against corruption. The article is organized as follows: the next section provides a short literature review of publications devoted to the methodological approach of economics to corruption. The third chapter analyzes mainstream economic approaches to corruption—

principal-agent theory and rent-seeking theory—and reveals their main problems. The fourth chapter shows how institutional economics and redistribution system theory can overcome the shortfalls of mainstream approaches. The following discussion concentrates on the issue of how the reduction in undesirable redistribution environments also leads to the reduction in corruption. The conclusion summarizes the main points and findings.

## 2. Literature Review

There are not many publications devoted to the methodological approach of economics to corruption, but it is possible to find some exceptions. This literature review provides a brief description of such texts. Some authors emphasize the importance of such research and what its output should be. Breit et al. (2015) note that theoretical studies of corruption can help to understand what corruption is, and which forms and dimensions it takes. Standard approaches to corruption need to be compared with alternative views, new opinions should be created, and a discussion should be conducted, as to which view of corruption is better able to justify, explain, describe, and analyze this phenomenon. It is not enough to apply theory to corruption, it is necessary to creatively theorize the phenomenon of corruption itself. This approach can revitalize the discussion of corruption as a complex phenomenon and different views can help to understand individual aspects of corruption (one aspect is relevant to some feature, another provides a view of another aspect). As the authors state (p. 327): " . . . critique needs to do more than simply challenging or polemicizing 'the mainstream' understandings. Critique is beneficial and productive only to the extent that something novel or alternative is generated—new theories or new approaches". Caiden (2010) similarly mentions that a theoretical view of corruption should remove obsolete and inappropriate knowledge, show what has proven to be successful, and integrate new thinking. The author explicitly writes (p. 10): "The virtue of theorizing is that it strives to reduce confusion, to simplify the evidence, to discard the obsolete and unverifiable, and to incorporate new thinking. In fast-changing times, it is important once in a while to pause and stand back to see what has been achieved and what still needs doing to overcome ignorance." From Caiden's point of view, corruption is an elusive topic inherent in human existence, as humans are imperfect beings. Therefore, it cannot be assumed that corruption will disappear completely. Theory must offer appropriate ways of analyzing the phenomenon, including awareness of the strengths and weaknesses of the individual analyses. However, Caiden (pp. 10–11) thinks that " . . . research into corruption is not exactly welcomed, encouraged, or supported, that theorists are held at arm's length, that their motives are suspected, and there may well be personal risks and repercussions if they delve too deeply and reveal too much about corrupt activities".

Theorizing about corruption often points to the problems of a specific approach to this phenomenon. Entman (2012) believes that the main contribution of critical analysis to corruption is debating what corruption is. He mentions that standard approaches overestimate some forms of corruption, but that such behavior does not even need to be corrupt. Standard approaches, in addition, do not pay enough attention to less usual forms of corruption, or they do not sufficiently discuss whether the phenomena that are not considered to be corrupt have some corruption potential. Sampson (2015) agrees with the opinion that although there is much more attention to corruption than before, the literature still neglects many phenomena that have or may have a corrupt character.

Some authors (e.g., Golden and Picci 2005; Beber and Scacco 2012) argue that theory should turn its attention to the complexities of corruption and abandon the binary view and division that concentrates only on clear situations (for instance corrupt/non-corrupt, good/bad, ethical/non-ethical). Corruption always occurs in a contextual environment. It is usually the consequence of many factors including individual traits, social norms and situations, the behavior of other people, and so on. It is often a symptom of other problems rather than the main problem. If a theory looks at corruption only as a dysfunctional behavior, it has a limited chance to understand the full range of its consequences and to identify or suggest possible drivers for change. This view is supported by findings in

behavioral economics: its experiments (many of them found no country variation, e.g., Philp 2006; Alhassan-Alolo 2007; Mazar et al. 2008; Charron et al. 2013) show that human behavior cannot be distinguished only as corrupt and non-corrupt, but it depends on context. An individual can act honestly and condemn corruption in one situation and participate in a corrupt contract in another one. For example, non-participation in a certain corrupt activity may be caused not only because of moral restraint but because such participation would not allow participating in another more advantageous corrupt activity. Hodgson and Jiang (2007) similarly doubt that corruption occurs only for private gains. They emphasize that there can be many other reasons why people engage in corruption. If theory and research do not embrace these and other factors their conclusions will be misleading with limited contributions to practice.

Several studies (e.g., Ledeneva et al. 2017) critically analyze the context in which the fight against corruption occurs. Ashforth et al. (2008) and Forsberg and Severinsson (2015) emphasize that a critical view must be applied to theories highlighting the harmfulness or duplicity of corruption as well as the practices applied by the so-called anti-corruption sector. These practices significantly contribute to how corruption is understood, what is covered by this concept, and what, on the other hand, is excluded. The authors further highlight that, for complete understanding, corruption must be studied on an individual, organizational, national, and even international level. Most studies, however, concentrate only on a specific part and neglect the circumstances in which corruption occurs. It must be emphasized that the fight against corruption is not only aimed to reduce corruption, it also has other motives—to support the agenda of anti-corruption organizations, or gain a more favorable position in international trade or relations, etc. Sík (2002) and Bratsis (2003), for instance, mention that one of the functions of the Corruption Perceptions Index is to eliminate obstacles to free capital flow and so help to make western countries richer, rather than reduce corruption.

To sum up the theoretical studies about corruption: they try to better define the essence of the phenomenon including the understanding of some ambiguity of the concept, to reveal the weakness of specific approaches to corruption, and to show that both corruption and the fight against it occur in specific contexts and often have hidden motives. However, from the point of view Pozsgai-Alvarez (2020), what prevails in the research now are quantitative studies that mostly provide passive commentary, resulting in an underdeveloped field as a whole. Our article tries to overcome this trend and considers what approaches to corruption should include.

## 3. Standard Economic Approaches to Corruption

Aidt (2016) mentions that economics mainly investigates corruption from two approaches:

1.  "The helping hand type of corruption: Corruption arises when a benevolent principal delegates decision-making power to a non-benevolent agent who commits corruption." (Aidt 2016, p. 147). The approach can be called the principal-agent theory (agency theory) of corruption.
2.  "The grabbing hand type of corruption: Corruption arises because non-benevolent government officials introduce inefficient policies to extract rents from the private sector." (Aidt 2016, p. 147). The approach can be called the rent-seeking theory of corruption.

A similar division can be found, e.g., in Muramatsu and Bianchi (2021), Fisman and Golden (2017), Lambsdorff (2007), and Otáhal (2007), and we use it in our analysis. We are aware that the reality is more complex and that some texts use a combination of both approaches or add another to one or both of them. However, to be able to fulfill our task, some simplification is necessary and this allows to determine the essential characteristics of the "economics of corruption". Although the article pays attention mainly to the weaknesses or problems of both theories, it must be emphasized that we do not consider them useless. Both undoubtedly contribute to the understanding of corrupt behavior and the

reasons for corruption. Many of their conclusions (e.g., that low accountability contributes to corruption) are valid. However, they must be included, as the text shows, in a wider context. The text does not analyze some other theories, for instance, transaction cost theory (e.g., Furubotn and Richter 2005), public choice theory (e.g., Tresch 2022), and theory of entrepreneurial allocation (e.g., Baumol 1990; Aeeni et al. 2019), as these can be, in our opinion, aligned with other approaches mentioned in the text (transaction cost to institutional economics, public choice to the agency theory, theory of entrepreneurial allocation to the rent-seeking theory). Some ideas of these theories are, at least indirectly, analyzed and mentioned in the following text.

*3.1. Principal-Agent Theory of Corruption*

3.1.1. The Essence of the Principal-Agent Approach to Corruption

The principal-agent theory of corruption is the most dominant approach of economics to corruption (Ugur and Dasgupta 2011; Marquette and Peiffer 2015; Prasad et al. 2019). Anti-corruption programs usually come from this approach (Villarreal-Diaz 2018). The theory sees corruption as a form of the principal-agent problem: the interests of a principal and an agent diverge. Due to asymmetric information when the agent has more information than the principal (Panda and Leepsa 2017), the principal is unable to perfectly monitor the actions of the agent, and so the agent can pursue its interests and behave against the principal's will, including participation in a corrupt contract that damages the principal's utility. Corruption differs from other forms of the principal-agent problem by requiring the presence of a third party, sometimes called a client (e.g., Groenendijk 1997; Szantó 1999; Fitzsimons 2009). The client colludes with the agent to create a corrupt contract in which the agent commits to breach the obligations defined by the principal in favor of the client or a person related to the client. The client or its related person somehow benefits from the agent's breach of duty.

The agent making the deal with the client damages the principal because the agent prefers the interests of the client and not the principal. Some authors (e.g., Shleifer and Vishny 1993) describe selected forms of corruption as corruption without theft. As an example, they highlight the situation in which an official sells a license for a government price plus a bribe. Such corruption from their point of view is seen as harmless—the government still receives its money. However, the bribe changes the conditions of the contract. Some license applicants may not be willing to pay a bribe and will not ask for the license, but they would without a bribe. The government loses income in this case.

Agency theory does not usually distinguish whether the agent actively looks for a corrupt contract or whether the contract is offered by the client and the agent only accepts it. Incentives are more important. If the situation allows (i.e., when the potential benefits outweigh the risk of getting caught) an agent commits corruption whenever possible. Corruption on an individual level is the result of a subjective evaluation of the opportunities for corruption, the profits resulting from corruption, and the cost of corruption. Klitgaard (1988), in this regard, describes corruption with the equation: corruption = monopoly + discretion − accountability. Posner (2001) suggests the solution to the problem by harmonizing principal and agent interests. If the principal's interests also become the agent's interests, the probability of the agent undertaking corruption or other forms of opportunistic behavior decreases. However, harmonization can be difficult or costly. It is further necessary to consider the client's interest in making a corrupt contract that can overwhelm the effort to harmonize. Successful harmonization must therefore discourage the client from participating in the contract. This can be achieved by increasing the client's costs, which include: 1. Costs of finding an agent; 2. Costs of the contract specification; 3. What a client gives to the agent (e.g., amount of bribe); 4. Other costs involved in the contract fulfillment; 5. Monitoring and prevention costs—the client must also monitor whether the agent does not breach the corrupt contract; and 6. Failure costs: they include the situation when (a) although the agent fulfills what it promised it does not succeed in satisfying the client´s needs, the client loses resources given to the agent; (b) the agent

does not fulfill what they promised to the client, but the client still receives what it wants from the agent; or (c) the agent does not fulfill what they promised to the client and client does not receive what it wants from the agent—the client needlessly uses its resources in situations (b) and (c). Agency theory pays little attention to these situations and to ways of increasing the uncertainty of the client whether the agent fulfills its promises. However, higher uncertainty and probability that the client loses or uses needlessly its resources reduce the client's willingness to commit a corrupt behavior.

### 3.1.2. The Problems of the Principal-Agent Approach to Corruption

Agency theory mainly expects that corruption concerns only agents and that principals do not participate in corrupt behaviors and have an interest in fighting against corruption. Or from a slightly different point of view (Martinsson 2021): the principal-agent framework requires a well-informed and active principal that supervises and sanctions corrupt dissonance from agents. These are strong assumptions or requirements. If corruption is widespread, there cannot be enough principals willing to curb corruption. Corruption can further be convenient for principals. Let us take the relationship between politicians (agents) and citizens (their principals) as an example. Politicians can corrupt citizens—offer them various advantages (e.g., social benefits, cheaper products). Citizens close their eyes to politicians' corruption for reciprocity and even vote for politicians who promise citizens some advantages. As early as 1977, it was argued (Rundquist et al. 1977) that voters may not vote against corrupt politicians because they engage in an implicit trade-off between corruption and policies. For instance, Fernández-Vázquez et al. (2016) provided empirical evidence in favor of this exchange argument between voters and politicians using data from the Spanish local elections and showed that the trading can be quite explicit when voters expect to receive direct side benefits from corruption (see also Cubel et al. 2022 for other details). Similarly, corrupt politicians as principals of government officials can tolerate their corruption or make deals with them about the division of corrupt spheres. Mungiu-Pippidi (2011) summarizes that many principals can act as patrons or gatekeepers for corruption.

In many cases, a person acts both as a principal and as an agent. Politicians can be mentioned as the typical example—they are both principals for government officials and agents of citizens. The fact that a person holds multiple roles often reduces his/her willingness as a principal to fight corruption as corruption is convenient for his/her agent role. The cost connected with the fight against corruption is another factor: too-high costs lower willingness to fight. Khaile et al. (2021) mention, for instance, unwillingness of principals (citizens and politicians) to reduce non-compliance in South African municipalities because the cost connected with the process of reduction exceeds, from the point of view of principals, their potential benefits. The quality of principals is affected by the environment in which they live, and it can be assumed that higher corruption reduces their ability and willingness to combat corruption (see, e.g., Baig et al. 2022 for specific details).

The anti-corruption procedures proposed by agency theory, such as the reduction in monopoly positions, higher transparency and accountability, changing executive preferences, or the creation of special anti-corruption agencies with the task of fighting and detecting corruption fail in the situation of organized and systemic corruption. If there are not enough subjects willing to reduce corruption, the space created by these procedures will not be filled by non-corrupt actors and it will continue to be used by subjects involved in corruption. It makes, for instance, little sense to use whistleblowing as a means of revealing corrupt behavior. If a whistleblower informs the public about such behavior, (s)he will probably face denunciation or contempt, and people participating in corruption cut down contact with him/her or even take action against him/her. Other ways of reducing the probability of committing a corrupt act are connected with similar doubts. Consider, as an example, the asymmetric design of the punishment of a corruption contract, when the corrupt agent is severely punished if (s)he breaches his/her obligation, but (s)he receives only mild punishment if (s)he accepts benefits from the corrupting agent without

other action. On the contrary, the corrupting agent is severely punished for offering a benefit to the corrupt agent but mildly punished if (s)he receives some benefits from the corrupt agent for breaching the duty (see, e.g., Lambsdorff 2007; Basu 2020; or Pramanik 2022 for other details concerning this issue including a laboratory experiment made by Dasgupta and Radoniqi (2021)). The asymmetric design, on the one hand, discourages the corrupt agent from breaching his/her duties which is his/her main crime, and on the other hand, discourages the corrupting agent from offering a corrupt benefit, i.e., the reason why the corrupt agent usually breaches his/her duty. In the situation of systemic corruption, when corruption infiltrates the bodies that should fight against it, however, there may not be a willingness to apply this design.

Similarly in the situation of systemic corruption, the reduction in monopoly power remains insufficient and in practice will not lead to higher competition. Increased transparency can cause those who are not involved in corruption to discover how widespread the corruption system is and all who benefit from corruption activities. Non-participants also find that they are losing if they do not deal with a corrupt contract and that it is more beneficial for them to be a part of a corrupt system than to fight against them. Anti-corruption agencies can reduce corruption only formally and they can be controlled by corrupt networks. E-governance in northern India can be mentioned as an example. Prasad et al. (2019) revealed that two different sets of accounts were kept by staff involved in the program—official online accounts and unofficial offline accounts that allow the official to accept bribes for their services. Sometimes corruption may not even be hidden. Louridas and Spinellis (2021) use the term "conspicuous corruption" (as an analogy to Veblen´s (Veblen 1899) "conspicuous consumption") for when somebody publicly commits a corrupt act or publicly informs people about its commitment to emphasize his/her status and position which permits such conduct.

The above can be summarized in the idea that the agency theory in its pure form does not pay enough attention to the context in which corruption occurs and the reasons why it occurs. The theory of collective action is sometimes (e.g., Mungiu-Pippidi 2011; Persson et al. 2013; Booth and Cammack 2013; Persson et al. 2018; Martinsson 2021) known as the remedy. The theory emphasizes the importance of how the other members of the system behave in deciding whether to engage in corrupt activities or to actively oppose them. If a decision-maker believes that most of his/her contemporaries behave corruptly, it is rational for him/her to behave corruptly too. Teorell (2007), with some exaggeration, notes that if almost everyone is involved in corrupt contracts, honesty is deviant behavior. Similarly, Uslaner and Rothstein (2012, p. 5) argue: "people in systemically corrupt settings participate in corrupt practices mostly because they perceive that most other agents play this game and that it thus makes little sense to be the only agent that acts honestly if one cannot trust others to be honest." Corruption networks force people to participate in illicit behavior if they want to stay in contact with their friends, colleagues, neighbors, or other people (an excellent example of such a network with low density, when most people did not know each other but they know that there is a high probability that a counterparty is the member of the network and accommodate their behavior to that fact, i.e., behave corruptly, can be found in Luna-Pla and Nicolás-Carlock 2020).

Let us mention that the importance of beliefs about the behavior of others is also analyzed by other authors who do not explicitly address the agency theory, e.g., Basu (2015, 2020) introduces a new approach (based on game theory) to economics and law and shows that the law will work only if it changes the beliefs about the behavior of others and outcomes. Whether an action will be in compliance or non-compliance with the law depends on beliefs about the behavior of others. Basu (2015) explicitly mentions that if each person chooses the same action as (s)he would have chosen in the absence of the law, each person must receive the same payoff as what (s)he would have received in the absence of the law. If all people expect that nobody abides by the law, expectations will be confirmed, and the law will not be abided by. Similarly, a law is not needed if it stipulates something that is already

observed. Basu (2020) points out the importance of social norms that can effectively solve many issues instead of laws.

It is, however, questionable whether the collective action approach really overcomes problems connected with the agency theory. As Marquette and Peiffer (2015, 2018) note, both theories expect individual rationality. According to agency theory, individuals decide whether to participate in corruption based on their revenue and cost calculations. The same is assumed by the theory of collective action that adds beliefs about how others behave as another factor affecting decisions. However, a decision is created even in a more complex context. Politicians, for instance, may not have to fight bureaucratic corruption because their mandate is too short, and they are unlikely to reach an efficient result in one election term. Their struggle may, on the contrary, reduce the quality of public services provided by bureaucrats. As a result, dissatisfied citizens will choose different politicians in elections.

The theory of collective action does not also assume the Impact of heuristics and cognitive distortions, i.e., behavior that is not based on a rational calculation of revenue and costs. Both theories come from the neoclassical economic paradigm, whereby each person has an exogenously given preference or utility function and takes decisions to maximize this. However, people do not always go by pure cost–benefit analysis (Posner 2000). Trust or persuasion of how other deals is an important factor in the decision to participate or not in corruption but must not be overestimated (Marquette and Peiffer 2019). The persuasion about the behavior of others can be easily biased or outweighed by other factors. The option of corruption may be, for instance, the most emotionally appealing option for the corrupting agent. A corrupt offer can further cause positive emotions in the corrupt agent, so (s)he will accept it regardless of the possible consequences (see also the end of Section 3.2.2 for other details).

The theory of collective action also neglects the idea that corruption helps to solve some critical issues in many cases. For example, while in developed countries social cohesion is maintained through public budget redistribution, in developing countries these budgets do not have sufficient resources to maintain stability. Through corruption, these resources can be secured. Walton (2013), as a concrete example, points to the situation in Papua New Guinea, where corruption is used to ensure the basic social standard for citizens who are excluded from social services. Gauri et al. (2011) concentrated on Honduras, a county hit by civil war. Their analysis shows that the inhabitants of Honduras perceive clientelist relations as an adequate way to achieve security in a highly violent and unstable environment. South Africa can be mentioned as the third example: only due to corruption can some people procure electricity or water and send their children to a school (Das 2015). There is no contradiction that people in a country where corruption is widespread usually condemn it (Rothstein and Torsello 2014; Prasad et al. 2019), but they participate in it. They have minimal other possibilities for surviving. Generally, corruption is not an innate behavior, but a behavior that results from many factors and circumstances. It is, for instance, affected by the rules of society, by needs that people must satisfy, by threats that people face and other factors. For instance, higher heterogeneity in a society increases corruption, especially in the case when people prefer to deal namely with the group they belong to. Corruption is used as a tool how to keep group unity and how to achieve an advantage in competition with other groups (see Cubel et al. 2022 and their lab experiment for details).

Agency theory does not adequately consider that each society can be hit by corrupt norms supporting agents´ corruption, that principals can be corruptible, or that their behavior can be influenced by the corrupt environment. The theory of collective action considers the context of shared beliefs about whether other members of the system are acting in a corrupt manner, but it looks at corruption only as a problem and it neglects that corruption is often an attempt by citizens who have no legal options to achieve their legitimate needs. The above-mentioned situations, when people have the legal right to receive some goods or services but do not obtain them without a bribe, are typical situations when people are forced to participate in a corruption deal.

If anti-corruption measures do not consider the context in which corruption occurs, and that can explain why it occurs, they will not be successful. Look, for instance, at the corruption in the health system. As Hutchinson et al. (2020) emphasize, the number of subjects here plays the role of principal—citizens, politicians, owners of health providers, and even health workers who may wish to call their managers or ministers to account (when, e.g., medicines are scarce and commodities do not arrive in facilities). The interests of each group of principals differ and it is difficult to harmonize them. If the interests of a group are not sufficiently represented, corruption can be used as a tool to achieve them. Employees may resort to corruption as a source of income if they are poorly paid. Similarly, patients may resort to corruption if other means of ensuring quality care fail. To combat corruption means to change this context to enable equal and quality health care for all citizens, not to try to punish those who participate in corruption.

Another problem is that anti-corruption effort sometimes tries to reduce corruption to a minimum level or eradicate it totally. However, if there is an awareness of universal principles of human behavior, common good, and fair public service, etc., there is always a possibility that these values will be violated. From this point of view, a society without corruption is an inhumane society—people would lose their human nature. Humans do not have the power and skills to prevent every violation of the universal rules of human conduct. Therefore, there will always be corruption or other forms of opportunistic behavior. Attempts to build social orders in which opportunistic behavior would not occur at all have always led to drastic restrictions on human freedom, and human suffering including the death of those who disagreed with such measures (the excellent example is Pol-Pot´s Campucia in the years 1975–1979 when the eradication of corruption, which was seen as the feature of urban life, was one of the aims of the tyrant's rules, see, e.g., Short 2007 for details). From this point of view, the anti-corruption movement must be approached critically and cautiously.

### 3.2. Rent-Seeking Theory of Corruption

3.2.1. Rent-Seeking Approaches to Corruption

The term rent-seeking will be understood in its common standard sense (e.g., Tullock et al. 2002; Congleton and Hillman 2015; Choi and Storr 2019): when resources are used to obtain rents coming from some activity that has negative social value. The term emphasizes that people may use not only productive but also non-productive activities in the form of gaining a privilege to earn their revenue. These activities do not increase the wealth of society but redistribute existing wealth in favor of privileged holders. Rent-seeking activities usually involve government, political decisions, or the public sector. Theory (e.g., Congleton and Hillman 2015) therefore concentrates on rent-seeking in these parts, rent-seeking that involves only private subjects is mostly peripheral.

The rent-seeking theory looks at corruption from two different views. The first and the prevailing one (e.g., Cartier-Bresson 1997; Lambsdorff 2002; Lambsdorff 2007; Chen 2010; Bjornskov 2012; Majeed 2014; Aidt 2016; Financial Express 2017; Choi and Storr 2019; Dinca et al. 2021; Laurent 2021; Khandan 2022) sees corruption as a form of rent-seeking. It can be compared with other forms, especially lobbying. The comparison now usually concludes that corruption is a more detrimental form of rent-seeking. This approach usually sees lobbying as a legitimate form of interest enforcement. It is argued (e.g., Zetter 2014) that lobbyists do not decide, they only try to affect decision-makers and present decision-makers with information. Some authors in the past (e.g., Wellisz and Findlay 1984; Appelbaum and Katz 1987; Kaufmann 1997) had the opposite opinion and thought that the effects of lobbying were, in comparison with the effects of corruption, more harmful. However, as Lambsdorff (2007) emphasized, corruption is usually secret, it can be made by one corrupting agent and thus it follows a narrower interest than lobbying. If a lobbyist represents an interest group, it must consider the interest of all its members, which is wider in scope than the interest of one person. If the interest group includes all members of society and promotes their interests, a lobbyist would advance the public interest. The open nature

of lobbying further forces lobbyists and persons influenced by them to keep their promises and commitments which reduces lobbying costs. The secretive nature of corruption means higher risks of a corrupt contract, which is connected with higher payments and additional costs. Lobbying can also bring some other positives. It sometimes concerns topics with limited attention, it can recruit new politicians, it can work as a means of socialization, and it helps shape citizens' attitudes and values. There are no such positives with corruption.

The second view, IIch is held especially by some authors connected with the so-called Austrian economic school (e.g., Rothbard 2011), considers rent-seeking itself to be corruption. Corruption, in this view, always arises when artificial barriers to entry to a market are created, even if the creation is made by legal means, and, because of the creation, the well-being of society is reduced. The view sees activities trying to eliminate or circumvent these artificial barriers as beneficial and legitimate even if they use tools such as bribery. The view further proposes (e.g., Meon and Weill 2010; Johnson et al. 2012) that corruption in the situation of over-regulation can smooth ("grease the wheels" of) entrepreneurial activities. To the contemporary studies finding positive effects of corruption belong, for instance, the paper of Cerdeira and Lourenço (2022), which concludes that corruption has a positive and statistically significant impact on innovation (based on data from 1062 Portuguese firms in the year 2019).

### 3.2.2. The Problems of the Rent-Seeking Approach to Corruption

Regardless of whether the rent-seeking theory perceives corruption as an undesirable form of rent-seeking, or whether rent-seeking itself is perceived as corruption, the theory usually recommends curbing government power to bestow privilege or to create barriers of entry, as the means of reducing corruption. The recommendation can be shortly characterized as "less government—less corruption" (e.g., Becker and Becker 1998; Tanzi 2000; Alesina and Angeletos 2005; Chen 2010; Hasen 2012; English 2014). Our article argues that this argument is flawed. It confuses the volume of corruption and the intensity of corruption. By simply reducing what the government decides without further steps, corruption can persist in those areas where government power remains. It can even increase here if it shifts there from the areas where government power was reduced. The removal or reduction does not necessarily mean the removal of the monopoly, corruption can stay here, it simply moves from the public to the private sector. The private sector creates room for corruption too. Private ownership is not just the owner's relationship to the subject of ownership. The relationship contains rules covering what the owner and others may and cannot do, including the protection and enforcement of the rules. If the rules, protection, and enforcement do not exist or they are only insufficient, reducing the number of public sector decisions does not resolve corruption or similar behavior. The private sector must be based on generalized trust and the market must support anonymous relationships and not prefer other criteria. Kruták (2010, p. 1) here clearly states: "There is little room left for new subjects and innovations in the environment of corruption networks. With a little bit of exaggeration, Henry Ford would have barely dominated his legendary "Tin Lizzie" with the US automotive market if all retailers refused to sell the car saying that he was cheaper, but that Chrysler paid them for a vacation."

The argument "less government—less corruption" also forgets that some privileges created by a government can be beneficial for market development. It does not answer the question of the definition of socially beneficial rules or barriers that do not encourage people towards corruption. Generally, rent theory, as well as agency theory, pays insufficient attention to the environment where corruption occurs. For instance, not only corruption but also other rent-seeking activities will be developed in the situation of systemic corruption, as such, an environment offers many opportunities to obtain rent. Firms collude secret contracts at this stage and divide a market among themselves. Lack of competition will lead to an oligopoly or monopoly market. Individuals and organized groups can then also use rent-seeking by gaining status or influence in these private entities—companies,

associations, and similar subjects. However, such a system generates high transaction costs. Corruption can then be used as a tool to overcome them.

The opinion that corruption can, in the situation of overregulation, "grease the wheels" of entrepreneurial activities is further flawed as it looks at corruption as an exogenous phenomenon. However, over-regulation can be created by officials and politicians to receive revenues from bribes. The willingness of some subjects to pay them is then seen as its weakness and these subjects are forced to provide additional bribes or benefits. Some authors (e.g., Ryska and Průša 2013) think that willingness to participate in corruption proves efficiency, as only the most efficient firms can afford to pay the bribes. However, the bribe can be offset by the lower quality of the firms´ products. As an example, in the situation when a construction company wins a tender due to a high bribe, it will save on the material, and the result is poor construction. Lambsdorff (2007) or Rose-Ackerman and Palifka (2016) mention cases where, due to corruption, some of the completed buildings were so poorly built that they collapsed. Others had non-functional fire equipment that failed to save lives when a fire broke out.

The consequences of corruption also have an impact on other areas that are not considered by the authors claiming that corruption increases efficiency, such as higher rates of child mortality and lower birth weight of children (both because of poor healthcare), higher environmental pollution, climate change (Leitão 2021), firms' innovation and performance (e.g., Ellis et al. 2020 or Huang and Yuan 2021 found a negative effect of corruption on innovation for US firms, Martins et al. (2020) investigated 21,250 firms in 117 countries for the period 2002–2016 and revealed that regardless of how performance is measured (sales growth, employment growth, productivity growth, or investment), firms that considered corruption as their main obstacle to business activities tended to obtain lower levels of performance).

Studies finding some positive effects of corruption (e.g., Cerdeira and Lourenço 2022) usually do not consider the longitudinal effect of corruption on the environment and human behavior. Although comprehensive evidence of Martins et al. (2020) indicates and confirms that corruption is beneficial from a second-best perspective, i.e., it can help to overcome bureaucratic constraints and rigid regulations and requirements, the main objection remains valid: corruption does not remove these obstacles, it only allows individual subjects, rather than all of society (i.e., each person), to overcome them and it creates incentives for officials to keep the existing regulations and requirements and to create others. The intention of politicians, officials, or other subjects (including firms benefiting from the barriers) to create and maintain barriers to entrepreneurial activities is an important context of corrupt behavior. Similarly, as in the case of agency theory, corruption must be seen in this context. The subject responsible for the creation of the barriers must be blamed first. Nevertheless, corruption is not an appropriate way to remove the barriers. As we mentioned in the introduction, corruption is a behavior contrary to universal principles and should not be preferred in solving the over-regulation problem.

Let us mention at the end of this section that the rent-seeking theory, like the agency theory, understands corruption as a rational action that people commit based on the calculation of benefits and costs and neglects some behavioral factors that can contribute to corruption. There is a list of some of them (own work, the titles of biases, heuristics or effects are taken from Ogaki and Tanaka 2017):

- Availability heuristic: Corruption may appear to be the most affordable option for both parties to the contract, they approach it without considering the consequences of their actions.
- Bias blind spot: One can assume that (s)he is not acting corruptly. While for others (s)he would understand the actions as corrupt, not for himself/herself.
- Confirmation bias: People can confirm their views on corruption and corrupt act, favoring sources that are in line with their views. Corruption thus becomes a state of inertia, even if non-corrupt action is more effective.

- Hindsight bias: Isolated acts of corruption can lead to the belief that the given system is affected by massive corruption. A reaction in the form of excessive anti-corruption measures can cause excessive bureaucracy, make human action (Hayek 2021) more difficult, and thus create space for further corruption.
- Dunning–Kruger effect: People can be wrongly convinced that nothing can happen to them in the event of corrupt behavior, that their corruption will never be detected. Thus, they commit it more often.
- Priming: If an individual is exposed to negative stimuli for a long time, (s)he may commit corruption in response to these stimuli. At the same time, corruption does not need to solve the given negatives, it can concern something else. The person acts corruptly just because (s)he is exposed to too many negatives.
- Non-monetary performance is perceived as less corrupt than monetary (Bandura et al. 1996): If the corrupting agent offers the corrupt agent a non-monetary performance, it is more likely that both subjects will not perceive the given transaction as a corrupt contract. Correspondingly, if they understand it as a corrupt one, it will appear to them less harmful than if money were given. Non-monetary performance is more easily interpreted as a gift for which it is appropriate to provide some corresponding consideration.
- Use of a middleperson: When a middleperson is used, it is not so obvious that the corrupt agent provides something to the corrupting agent and vice versa. The bond between them is weaker, the whole transaction can appear as a standard contract.
- Inducing a feeling of a debt, etc., in the counterparty of the contract. Thanks to the given feeling, the counterparty will be more willing to participate in a corrupt contract that will be seen as a form of the debt payment.

**4. Ways to Overcome the Shortcomings of Standard Economic Approaches to Corruption**

*4.1. Institutional Economics and Corruption*

4.1.1. Extractive Institutions as the Source of Corruption

As we summarize, neither the rent-seeking theory nor the agency theory pay sufficient attention to the environment or context of when and why corruption occurs. How can the context be incorporated into both theories? (Our aim is not to completely abandon both theories but to extend them by other relevant factors.) One possibility includes institutional economics. It emphasizes that corruption, especially in the form of organized or systemic corruption, is the consequence of the institutional design (structure of institutions) of a society. Institutions are (North 1991a: p. 97): "humanly devised constraints that structure political, economic and social interactions". It can be divided (Furubotn and Richter 2005; Groenewegen et al. 2010) into formal rules given by constitutions or laws and informal restraints given by taboos, customs, traditions, codes of conduct, and other informal rules. Their general aim is to keep order and safety within a market or society. Institutions can be seen as rules for human behavior that are the product of intentional (formal institutions) and non-intentional (informal institutions) human behavior.

Corruption occurs if an institutional design allows it, if people see it as acceptable behavior or if persons that should punish it abandon or do not fulfill their tasks. As Villarreal-Diaz (2018) argues, it is reasonable to recognize that people are not individuals with perfectly virtuous characters (at least all the time). The structure of institutions in a certain society affects the decisions of humans living there including their costs and yields connected with participation in corrupt activity. The theory should therefore ask which institutions support corruption and which ones reduce it.

Institutional economics surprisingly does not give clear answers to this question. Comprehensive publications about institutional economics (e.g., Furubotn and Richter 2005; Groenewegen et al. 2010; Kasper et al. 2013; Voigt 2019) do not pay detailed attention to corruption. Following Acemoglu and Robinson (2012), our article divides institutions into inclusive and extractive, economic and political (see Table 1) entities, and it argues that corruption is the product of extractive institutions. These institutions guarantee

privileges, monopoly, or oligopoly positions to some subjects, they reduce innovation and the process of entrepreneurial discovery. Non-holders of the privileges use corruption to obtain some of them or to be able to manage their business. Holders use corruption as a form of protection or expansion in the struggle for other privileges. Persons with the highest position and the greatest privileges can tolerate corruption below them, such as bureaucratic corruption. Additionally, officials tolerate and do not object to political corruption because it allows them to commit their corruption. Different forms of corruption coexist in a mutual symbiosis and help each other. In general, extractive institutions exclude or limit some individuals from access to economic or political activities. The excluded cannot develop and use their abilities. On the other hand, extractive institutions guarantee some people almost absolute power. We are reminded of Lord Acton's dictum (see, e.g., Himmelfarb 2015 for details): "Power tends to corrupt, and absolute power corrupts absolutely." From the opposite view, it can be stated that inclusive institutions support strong political capacity that leads to lower levels of corruption (an empirical confirmation can be found, e.g., in Baig et al. (2022)).

**Table 1.** Division and short characteristics of institutions.

|  | **Politic** | **Economic** |
|---|---|---|
| Inclusive | They provide enough plurality and contain the desirable degree of centralization—there is an authority in the society protecting human and property rights, voluntary contracts, and free entry to the markets. | They enable people to dedicate their lives to the kind of profession that best suits their talents. They enforce property rights, create conditions for equality of opportunities, and encourage investment in new technologies and skills. |
| Extractive | Opposite of inclusive. | Opposite of inclusive. |

Source: own creation based on Acemoglu and Robinson (2012).

The plausible impact of extractive institutions on the level of corruption is schematically drawn in Figure 1. There is always some level of corruption (its minimum value is represented by A) even if the impact of extractive institutions (their amount and how much they affect human relations and behavior) is small—e.g., Transparency International shows in its Corruption Perception Index (CPI) that there is no country without corruption, i.e., which achieves the index value 100 (Transparency International 2021b). At point B, when extractive institutions reach the minimum threshold corruption begins to grow but its growth rate is smaller than the growth rate of extractive institutions until point C. At point C, extractive institutions are widespread and affect most human activities. Corruption is commonly used in this situation not only to protect or extend the position of those who profit from extractive institutions but also as a means of improving the standard of living or of achieving legitimate goals of other people. Therefore, it grows at a higher rate than the level of extractive institutions. If an actor chooses to perform the virtuous act (not participating in corruption) in an environment where extractive institutions prevail it imposes unreasonable burdens on itself without clear benefits. We emphasize that the exact value of points A, B and C depends on the specific country's situation, i.e., there is no universal (world) value for all countries. The value can be affected by the moral awareness or ethical education of residents of a country, geographical factors, how often contacts and relationships between residents occur, how well the organizations fighting against corruption (e.g., police court, anti-corruption agencies) work, etc.

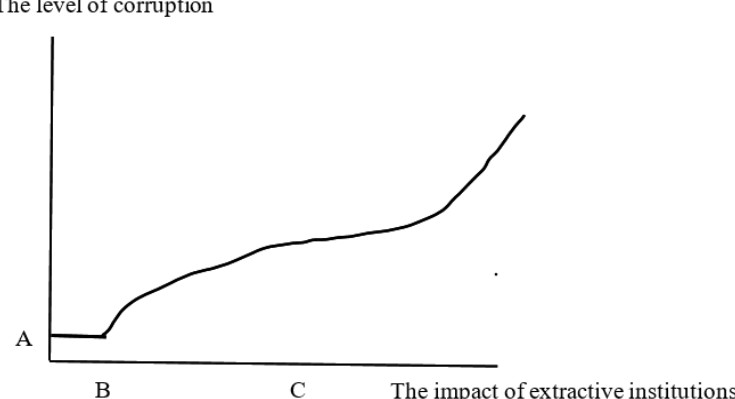

**Figure 1.** The relationship between the impact of extractive institutions and the level of corruption. Source: own creation.

4.1.2. Factors Supporting Extractive Institutions and Preventing Institutional Changes

If extractive institutions support corruption, why they are not removed? Each institutional change is costly and contingent: it must proceed from or respond to existing institutional design. Institutional economics (e.g., Groenewegen et al. 2010) uses the term "path dependency" that emphasizes dependency on the previous development. If there were institutions in a world of zero transaction costs, history would not matter. Changes in relative prices and preferences would prompt immediate restructuring of institutions to be more effective. However, a world with positive transaction costs means that inefficient institutions remain in existence for a long time even if conditions change—their change was too small and unprofitable to bring institutional change into effect.

Informal institutions in particular change slowly. The slow pace of change in some institutions is economically, evolutionarily, and otherwise advantageous—the stability of these institutions gives people confidence and allows them to anticipate the behavior of others. People are not forced to invest time and other scarce resources into constant change. If a system continuously changed all its parts, it would have the character of perfect chaos—the cost of adapting to changes would be greater than the benefit of the adaptation. The cost of adaptation will also be, due to permanent change, spent unnecessarily—adoption to some conditions makes no sense if the conditions immediately change. In such a system, it would not even be possible to cooperate, participate in the division of labor, innovate, and use many other attributes that help improve system performance. The convenience of slow institutional change, however, results in the persistence of ineffective economic institutions. Path dependency is supported by other factors. From our point of view, the most essentials include the following:

- Because of the logical structure of their mind (Kahneman 2012), people can overestimate the cost of an activity (costs are usually related to the present and the immediate future) and underestimate the benefits of this activity (the benefits will only manifest in a rather distant future). In this case, they may not be motivated to change inefficient institutions and create more effective institutions.
- If the benefits of institutional change will also apply to persons who are not involved in the change (free riders), people providing and implementing the change will probably achieve insufficient yields and their interest in realizing the change disappears.
- The consequences of inefficient institutions will usually occur with some time lag. During the lag part of the system, however, members of the system become accustomed to the institutions, and any change will be too costly for them—they would have to change their mental behavior patterns and models (North 1991b). It must be emphasized, from the point of view of human capital theory (Becker 1993), that it may not be convenient for those persons previously investing in their human capital, including accommodation to the existing institutional design, to change their

human capital structure, e.g., due to their life expectancy. If these people have a leading position in the system (which is quite likely), the effort to change inefficient institutions can be hindered, at least for some time. More generally, if the consequences of inefficient institutions manifest themselves with a time lag, the consequences need not (at least for some time) be associated with ineffective institutions.

- The human mind can be burdened with prejudices, which are a product of long-term development. People can prefer certain institutions, although these institutions may not be optimal to achieve their goals. The costs of changing the prejudices along with the cost of changing the institutions may be too large, so subjects are not willing to invest in an institutional change. Existing prejudices may even cause people to not even think about a possible institutional change.

If theory and practice want to remove extractive economic and political institutions and so reduce corruption, they must take all the above-mentioned factors into account. They must investigate and characterize the environment in which corruption occurs. Redistribution system theory serves here as a useful tool for analysis.

*4.2. Redistribution System Theory and Its Approach to Corruption*

4.2.1. The Essence of Redistribution System Theory

Redistribution system theory (e.g., Budínský and Valenčík 2009) investigates the causes, manners, and consequences of redistribution. Redistribution is defined as the situation when a subject is rewarded differently for its performance than that based on the person's productivity and the demand for their activities. Redistribution can be performed directly when some authority (either formal or informal one) decides that revenues or resources that belong to some person will be removed from it and they will be given to another subject, or indirectly when some barriers prevent a person from using all their resources and skills. The theory (e.g., Valenčík and Wawrosz 2014) emphasizes that some forms of redistribution are desirable—these are made to support or protect children, the sick, the elderly, and other subjects who are not able to at least partially provide the means for their livelihood. From that point of view, people have always lived in a redistribution environment that is framed by formal and informal institutions. Considering other viewpoints, redistribution research is connected with institutional economics (for details, see, e.g., Padovano et al. 2021). Desirable redistribution can, however, justify another redistribution form when a beneficiary of the redistribution can provide a sufficient amount of resources and goods for its livelihood without redistribution. Such redistribution is called (e.g., Browning 1978; Smith and Webb 2001; Shavell 2003) undesirable and it is defined in our article as the redistribution that does not satisfy Rawls' principles of justice (e.g., Rawls 1999, p. 266). It does not support situations when:

- "Each person is to have an equal right to the most extensive total system of equal basic liberties compatible with a similar system of liberty for all.
- Social and economic inequalities are to be arranged so that they are both: (a) to the greatest benefit for the least advantaged and (b) attached to offices and positions open to all under conditions of fair equality of opportunity."

Incentives supporting foreign investments, especially in the form of tax relief or government subsidy, are typical examples of undesirable redistributions. Some firms are favored in comparison with others. The favored firm can easily achieve higher revenue and profit. As OECD (2002) emphasizes, if many countries use such favors and compete for foreign investors, such competition may lead governments to engage in "bidding wars" that drive up investment subsidies to exorbitant levels, and/or drive down public measures that are needed to protect the environment and/or workers' rights and labor standards. Some studies (e.g., Tavares-Lehmann et al. 2016) already found that the most profitable firms receive the incentives, although they would invest in the county providing incentives even in the situation of no incentive. Firms consciously say that they would not invest without an incentive, although they would. Generally, although it can be argued that the redistribution theory does not give a clear clue as to what exactly undesirable

redistribution means or everything it includes, key philosophers including Aristotle (e.g., Aristotle 2009, 2017) Adam Smith (e.g., Smith 2011) Immanuel Kant (e.g., Kant 2017) and John Rawls (Rawls 1999) at least indirectly show that a human is able to distinguish whether an individual redistribution case belongs to its desirable or undesirable form.

The redistribution system theory investigates the conditions under which is redistribution (both desirable and undesirable) provided. An individual subject usually does not have enough power to create the situation when it is redistributed in its favor. It must create a collation with other subjects. The coalition then strives for redistribution in favor of all members of the coalition and at the expense of nonmembers. Nonmembers can, of course, prevent such development. Let´s imagine a situation of zero transaction costs and the same bargaining power of all members of the society. The effort to protect his/herself against redistribution would lead, under such conditions, to the situation when discriminating coalitions (redistribution is done in their favor) would arise purely randomly and the amount that a member of the discriminating coalition would receive in addition to his/her remuneration in the situation without redistribution would be the same for all coalitions (i.e., regardless of which coalition a person is a member of). Because these conditions are not met, institutional design determines and affects who will be a member of the discriminating coalition.

### 4.2.2. Undesirable Redistribution and Corruption

Several authors (e.g., Haller 2008; Bělohradský 2011; Agbiboa 2012; Yue and Peters 2015; Monteverde 2021) ask whether some, albeit legal, forms of behavior (such as investment incentives, above-standard state-business relations) have more damaging consequences than corruption. The political connections of firms with politicians can be mentioned as a typical example. CEOs of such firms use firm resources to help connected politicians. It is questionable whether the help is convenient for a firm (when for instance the firm acquires public procurement thanks to it). The results are ambiguous. As Bertrand et al. (2018, p. 850) mention, "CEOs might have personal benefits from a continued relationship with a politician or might find it difficult to ignore pressures from politicians that belong to the same political network." The authors found (the study is based on French firms' data) (p. 874) that accounting performance at firms managed by connected CEOs is lower than non-connected firms and decreases as the fraction of plants that are located in contested areas increases. The lower performance is further driven (primarily) by higher labor costs. However, a similar study concerning China (Fan 2021), where the law prohibited government officials from serving on corporate boards and receiving any income from firms, found that the returns of the firms declined after the resignation. The more politicians in the board of directors, the bigger the decline. Palanský (2021) similarly found that Czech firms (data between 1995 and 2014) that had donated to a Czech political party outperformed their non-connected but otherwise similar competitors by 8–12% in profitability, but higher profitability was also connected with other forms of connections—for instance, in the case firms that had filled public procurement contracts or had received public grants.

What the above-mentioned and other forms have in common with corruption is that they belong to undesirable redistribution. If these forms are legal, they take place in the public sphere. Redistribution system theory (e.g., Valenčík and Wawrosz 2014; Otáhal et al. 2013; Wawrosz 2019) therefore emphasizes that it is not possible to analyze corruption in isolation. It is necessary to investigate and concentrate on the environment in which corruption occurs. If undesirable redistribution spreads in a society, it creates conditions by increasing inequality and lowering generalized trust (e.g., Rothstein and Uslaner 2005) for the expansion of corruption as a form of undesirable redistribution that usually has (in democratic countries) a secret and hidden character.

Empirical relationships between corruption and income inequality show for instance that:

- an increase in corruption by one point (measured by CPI) reduced income for the poor by about 7.8% (Gupta et al. 2002, cross-country regression analysis for 1980 to 1997).

- an increase in one standard deviation in corruption (measured by CPI) means an increase in income equality by approximately 3.8 percentage points (Sánchez and Goda 2018, sample of 148 developing and developed countries over the period 2003–2015).

Current development further reveals that while inequality among countries is currently narrowing, inequality within developed countries rises. According to Oxfam (2016) or United Nations (2020), inequality is the consequence of the system allowing the rich to become richer and preventing the majority of the population from any increase in prosperity. Higher-income inequality is a sign of undesirable redistribution. The rich use their privileges to increase their position and to prevent others from earning enough money for their living. An example of the privileges can be, according to Stiglitz (2012), transfers and subsidies from the government to the rich, regulations resulting in less-competitive markets, insufficient enforcement of existing competition laws, inconsistent implementation of laws and norms allowing corporations to take advantage of others or to pass costs on to the rest of the society.

Undesirable redistribution resulting in income inequality and corruption are interconnected. Higher-income inequality means that people in the country do not have equal opportunities and chances. Some of them obtain their income and wealth due to privileges that they are endowed with; others are prevented from developing and using their skills. Inequality then fosters corruption. In some cases, inequality intentionally motivates an individual or group to commit corrupt behavior to protect their privileges. In other cases, inequality happens to be a factor that in different ways facilitates certain forms of corruption.

It can be objected that the redistribution system theory has some common features with the rent-seeking theory and that undesirable redistribution can be seen as the consequence of rent-seeking. Why another term with a similar meaning should be introduced? We argue that undesirable redistribution can occur not only due to this consequence but also due to other reasons. It may, for instance, have a longitudinal tradition, people may take it for granted and there is no necessity for the holders of a privilege to take any action to obtain or maintain it. Redistribution is supported by informal institutions, by beliefs of the members of the society. Sometimes a person receives a privilege, although (s)he does not endeavor to, and simply the fact that (s)he holds it is beneficial for other subjects. Generally, the rent-seeking view can be too narrow for the analysis of all cases of undesirable redistribution.

## 5. Discussion

An undesirable redistribution environment can be seen as an environment supporting corruption. To reduce corruption thus means not only concentrating on its obvious forms but also on all forms of undesirable redistribution. A society without undesirable redistribution guarantees equal opportunities and allows its members to pursue and realize their goals and ideas. It also ensures that the rights of others are not violated. Such a society is the most integrated and complex. Its social and economic transactions take place between all groups or strata. Compliance with what has been agreed upon in these voluntary transactions is guaranteed:

- both through legal rules and compliance organizations (e.g., courts),
- and by informal institutions (type of business practices) and voluntary associations, to which all or at least a majority of society members have access.

Such a society generally strives for the protection of property rights, voluntary contracts, and the most open access to individual sectors. Politicians in the society aim to achieve the general good, reconcile interests, or peaceful, non-violent, and consensual solutions to the conflicts between individual subjects in the society. Political power is a means to achieve these goals and not a goal in itself. It is limited both formally and informally. Formal constraints consist of defined mechanisms, what to decide, and how. The informal constraints are based on the fact that the people are involved in political decision-making; individuals realize that they are not entirely autonomous beings, rather

that they must take into account in their decisions the legitimate interests of others. There is a desirable level of centralization that enables effective decision-making. Institutions in the society are inclusive economic and political institutions (see Section 4.1) preventing the spread of corruption.

If the above-mentioned happens, the performance of agents will be more transparent, and principals will be able to create motivation systems harmonizing principals' and agents' interests more easily. A society without undesirable redistribution easily reveals corruption or other opportunistic behaviors of principals. People will not need to solve the dilemma of when others participate in corruption. The fight against corruption will be realized with lower costs. Redistribution system theory, with its focus on the redistribution environment and division of redistribution into desirable and undesirable can thus overcome the shortfalls of the agency theory and collective action theory when it reveals the characteristic of the society when (mainly excessive or systemic) corruption happens. It can also overcome the shortfalls of the rent-seeking theory. It answers the question of which barriers to entry or privileges should be removed—barriers and privileges resulting in undesirable redistribution—and which should be preserved—the remaining. Although it will probably be impossible to reduce all forms of undesirable redistribution, both theory and practice must pay attention to its forms and ways in which to reduce them. If theory concentrates only on corruption, other forms of undesirable behavior with more serious consequences may stay hidden and the causes of corruption will be eliminated.

## 6. Conclusions

Economics usually researches corruption from the point of view of principal-agent theory and rent-seeking theory. However, both suffer some shortfalls because they do not pay enough attention to the environment where corruption occurs. The principal-agent theory further expects a willingness of principals to fight against corruption and does not assume that they can tolerate it or even participate in it because some of its forms are convenient for them. The theory thus gives no clear answer as to how to stimulate non-corrupt behavior in agents. The rent-seeking theory does not sufficiently distinguish which barriers to entry and privileges are socially effective and improve the well-being of society. Its recommendation for curbing the government's power to provide privileges confuses the volume of corruption and the intensity of corruption. By simply reducing what the government decides without further steps, corruption can persist or even increase in those areas where government power remains, and it can move from the public to the private sector.

Both theories should be extended by other approaches. The article mainly analyzes institutional economics and the theory of redistribution systems. It only partially mentions theories such as behavioral economics and we emphasize that their ideas, e.g., the division of human thinking into fast and slow (Kahneman 2012), behavioral biases and heuristics (e.g., Ogaki and Tanaka 2017), also offer an interesting insight into why people commit corruption acts and should be also incorporated in economic analysis of corruption. Institutional economics sees corruption as the consequence of institutional design (structure of formal and informal institutions in society). It, however, does not answer the question of which institutions contribute to corruption. This article deduces that they are extractive economic and political institutions. Neither allow citizens to develop their skills and devote their abilities to those areas with lower opportunity costs. To reduce corruption means changing extractive institutions to inclusive ones. However, such change can be costly or hindered by many factors mentioned in the article. Corruption is a form of undesirable redistribution that is researched through the redistribution system theory. If a society succeeds in reducing the forms of undesirable redistribution, it creates space for harmonizing principals' and agents' interests and for the identification of which barriers to entry and privileges should be removed, and which ones should be preserved. Thus, the theory must pay attention to all forms of undesirable redistribution and not concentrate only on corruption, as the other forms create space for the spread of corruption.

**Funding:** This research received no external funding.

**Institutional Review Board Statement:** Not applicable.

**Informed Consent Statement:** Not applicable.

**Data Availability Statement:** Not applicable.

**Conflicts of Interest:** The author declares no conflict of interest.

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
