# Peer review of "How Corruption Is and Should Be Investigated by Economic Theory"

_economies, doi:10.3390/economies10120326_

Round 1
Reviewer 1 Report
This paper is a thorough literature review that discusses the corruption issue under the principal-agent, rent-seeking and institutional economics framework. I think the flow is clear and generally decent. I just have the following suggestions listed below.
1. One of the key is, WHAT IS corruption. In my opintion, a vast literature discusses corruption issue under different frameworks is because they are not reffering to the same thing. For example, whether it is for private interests, whether it harms public interests, whether it is illegal, can give different forms of the so-called corruption. The author(s) also mentioned this point at the beginning of the paper. I would expect a deeper discussion on how the definition of corruption may affect people's choice of research framework.
2. A vast of literature may discuss political connections (for example, see references). What is the relation between political connections and corruptions?
3. While corruption is generally undesired, what is the negative impact of anti-corruption (if any)? The literature in 2. sometimes mentions the information transmission and/or correlated equilibrium as a good side-effect of corruption. The author may add a discussion whether the anti- corruption campaigns bring unwanted result, though.
Bertrand, M., Kramarz, F., Schoar, A., & Thesmar, D. (2018). The cost of political connections. Review of Finance, 22(3), 849-876.
Fan, J. (2021). The effect of regulating political connections: Evidence from China's board of directors ban. Journal of Comparative Economics, 49(2), 553-578.
Fisman, R. (2001). Estimating the value of political connections. American economic review, 91(4), 1095-1102.
Author Response
Thanks for your valuable comment.
I add in Introduction a definition and characteristic of corruption and I emphasize that each definition or characteristic frames and limits the scope of the research.
The issue of political connections is mentioned in the first paragraph of chapter 4.2.2 and political connection is seen as the example of undesirable redistribution.
The problems connected with anti-corruption campaign are discussed in the last paragraph of chapter 3.1.2.
The author.
Reviewer 2 Report
The article offers an interesting reflection on how corruption is and should be investigated in economics. This said, I believe the present manuscript requires substantial revisions before it should be published.
The introduction does not clarify what the contribution of this paper is and to what literature. The paper discusses a few theories, finds some of them wanting, and advocates a couple of others. But this needs to be better sold in the introduction, not only by emphasizing the sort of conclusions the paper aims at, but by identifying the potential interlocutors who would be interested in this discussion.
The section “Literature Review” is fine, but should be revised with a view to showing how the several papers quoted articulate with each other. There is a bit of a “X said this, Y said that, Z said that” feel to it that limits the impact of the section going forward.
Section 3 begins with “it can be generally concluded”, but the authors offer no arguments or references to defend this claim. Regarding the authors’ take on principal-agent theory, this reader is far from persuaded by the claim on pages 5-6 that principals qua principals also participate in corrupt behavior. There is something amiss here. In the examples given the principle is actually an agent of someone else, and the corrupt behavior happens within this relationship. The appearance of “the theory of collective action” was also a bit sudden. In all, I believe some of the criticisms raised against both the principal-agent theory and the rent-seeking theory are fine, but their impact is not clearly fleshed out. After all, every theory is limited. In other words, the authors should clearly distinguish failures from limitations of those theories and emphasize the importance of each in our understanding of corruption.
It was also unclear to me the extent to which the insights of these two theories are incompatible with those that the authors defend. I believe the authors should better articulate these relations. Are not these theories better seen as complementary? If not, why not? If yes, how?
Apart from these comments, I would just further recommend a thorough proofing of the manuscript, as there are occasional typos (the abstract is rife with them).
Author Response
Thanks for your valuable comments.
I specify in Introduction what the main contribution of the article is, from my point of view. The article also emphasizes that the traditional approaches to corruption are not useless. Both undoubtedly contributed to the understanding of corrupt behavior and the reasons for corruption. Many of their conclusions (e.g., that low accountability contributes to corruption) are valid. However, they must be included, as the text shows, in a wider context.
Literature review are newly organized and each paragraph now specifies what specific texts dealing with methodological issues concerning corruption research emphasize, what they are concentrating on. The end of the chapter summarizes the main topics that are discussed in the reviewed articles.
The beginning of third chapter (between title 3 and title 3.1) is rewritten and extended, it is explained why the agency theory and theory of collective action are investigated. The text also emphasize the importance of both theories and that the critique does not want to abandon them bud add other approaches and views to both theories. Chapter 3.1.2 now clearly specify (in the second paragraph) that one problem connected with the agency theory is that a person can act both as a principal and an agent. Chapters also summarize that agency theory in its pure form does not pay enough attention to the context in which corruption occurs and the reasons why it occurs and that the theory of collective action concentrates on one of the factor affecting human decision whether to participate in a corruption contract, namely how the other members of the system behave, i.e., if they participate or not in such contracts.
Chapter 4 tries to better explain what institutional economics and redistribution system theory can add to both theories.
The author.
Reviewer 3 Report
This article discusses the economic approaches to corruption and how this phenomenon has been analyzed in the literature (and how it should have been analyzed).
As for the paper, I have the following comments:
- As for the purpose of the paper (section 1 – introduction), the authors states that he wants to “show how standard theories, specifically agency theory and rent-seeking theory research corruption, to reveal problems connected with the theories and to show how alternative approaches, mainly institutional economics and redistribution system theory, can overcome problems connected with standard theories.” One wonders if the author considers that these two “standard” theories are useless to explain the corruption phenomenon or whether they can complement each other and other theories. As corruption is a complex phenomenon (which is something the author recognizes), one would expect that a single theory would explain this phenomenon in an imperfect way (at best) and that several theories should be brought together to explain such a complex phenomenon as corruption. However, nothing is said about the merits and contributions of the standard theories.
- Section 2 is supposed to be about a “short literature review of publications devoted to the methodological approach of economics to corruption”. However, the author does not present the methodological approaches to corruption; instead, the section goes on listing some authors that complain about how corruption has been dealt with by the economics literature. While this may be relevant, the methodological contributes should be described/explained in this section and they are not.
- As for section 3, the author describes two theories: principal-agent and rent-seeking. While these theories are relevant to explain corruption, I fail to understand why others (e.g., transaction cost theory, public choice theory, theory of entrepreneurial allocation – See, e.g., Martins el al., 2020 and Seck, 2020) have been left out of the discussion. While the author may choose to focus his analysis on the two selected theories, at least there should be a brief explanation on why these two (and not others) were selected.
- Section 3.1.2 on the problems of the agent-principal theory make me uncomfortable. This is because the main argument of the author is the possibility of the principal, as the agent, may want to engage in corruptive activities. However, the examples brought (namely, considering politicians and citizens), however interesting, seem to contradict the context under which the principal agent theory is founded. In these examples, the principal becomes another agent (and therefore, it is an agent of other (who?) principal), so the theory continues to apply. I would suggest that the author explains/reformulate the argument so that it becomes intelligible.
- In section 3.2.2., the author suggests that “corruption should not be used as a way in which to solve over-regulation.”. This is related with the so-called “greasing the wheels” hypothesis, under which corruption allows firms to overcome excessive, bureaucratic regulations. While the literature has embraced both the “sands of the wheels” and “greases the wheels” hypothesis (which, by the way, should be discussed here in this paper in a more thorough way), the greasing the wheels hypothesis considers that firms have the incentive to corrupt to overcome overregulation. Corruption is not a solution, it simply is the result of the incentives that firms face in such a disadvantageous institutional context. But is may be the kind of “second-best” solution that firms find to “get thing done”. I suggest the author adjusts the argument presented in this section considering these elements.
- Please check the text for typos as there are a few of them (e.g., the title of section 3.1. should be “agent” and not “agnet”)
References:
Martins, L., Cerdeira, J., Teixeira, A. (2020). Does corruption boost or harm firms’ performance in developing and emerging economies? A firm-level study. The World Economy, 43(8), 2119-2152
Seck, A. (2020). Heterogeneous bribe payments and firms’ performance in developing countries. Journal of African Business, 21(1), 42–61. https://doi.org/10.1080/15228 916.2019.1587806
Author Response
Thanks for your valuable comments.
I specify in Introduction what the main contribution of the article is, from my point of view. The article also emphasizes that the traditional approaches to corruption are not useless. Both undoubtedly contributed to the understanding of corrupt behavior and the reasons for corruption. Many of their conclusions (e.g., that low accountability contributes to corruption) are valid. However, they must be included, as the text shows, in a wider context. I also add in Introduction a definition and characteristic of corruption and I emphasize that each definition or characteristic frames and limits the scope of the research.
Literature review are newly organized and each paragraph now specifies what specific texts dealing with methodological issues concerning corruption research emphasize, what they are concentrating on. The end of the chapter summarizes the main topics that are discussed in the reviewed articles.
Literature review are newly organized and each paragraph now specifies what specific texts dealing with methodological issues concerning corruption research emphasize, what they are concentrating on. The end of the chapter summarizes the main topics that are discussed in the reviewed articles. It is explained in the beginning of third chapter (between title 3 and title 3.1why the text does not deal with other theories (e.g., transaction-cost theory). Some ideas of these theories are, at least indirectly analyzed in mentioned in the text – e.g. analysis why ineffective (extractive) institutions prevail mention that an institutional change is costly and this cost includes also transaction cost.
Chapter 3.1.2 now clearly specify (in the second paragraph) that one problem connected with the agency theory is that a person can act both as a principal and an agent. There are also other changes both in chapter 3.1 and 3.2 that, as I hope, better analyzes the shortfalls of the agency theory and principal agent theory. Section 3.2.2 concerning the “greasing the wheels” hypothesis shows that corruption can be used as the second best solution of overregulation but it does not remove these obstacles, it only allows individual subjects and no to all society (i.e. to each person) to overcome them and it creates incentives for officials to keep the existing regulations and requirements and to create other. The intention of politicians, officials, or other subjects (including firms benefiting from the barriers) to create and maintain barriers to entrepreneurial activities is an important context of corrupt behavior. Chapter 4 tries to better explain what institutional economics and redistribution system theory can add to both theories.
The author.
Reviewer 4 Report
In general, the paper suffers from a severe lack of clarity, regarding why this investigation is interesting, what the main contribution is relative to the literature, how the variables are defined, and what the results tell us. In Figure 1. it is not clear how the author obtains the relationship between the impact of extractive institutions and the level of corruption. The work does not seem to fit the requirements of a reputable journal and requires a more solid methodological background
Author Response
Thanks for your valuable comments.
I specify in Introduction what the main contribution of the article is, from my point of view. The article also emphasizes that the traditional approaches to corruption are not useless. Both undoubtedly contributed to the understanding of corrupt behavior and the reasons for corruption. Many of their conclusions (e.g., that low accountability contributes to corruption) are valid. However, they must be included, as the text shows, in a wider context. I also add in Introduction a definition and characteristic of corruption and I emphasize that each definition or characteristic frames and limits the scope of the research.
Literature review are newly organized and each paragraph now specifies what specific texts dealing with methodological issues concerning corruption research emphasize, what they are concentrating on. The end of the chapter summarizes the main topics that are discussed in the reviewed articles.
Literature review are newly organized and each paragraph now specifies what specific texts dealing with methodological issues concerning corruption research emphasize, what they are concentrating on. The end of the chapter summarizes the main topics that are discussed in the reviewed articles. It is explained in the beginning of third chapter (between title 3 and title 3.1why the text does not deal with other theories (e.g., transaction-cost theory). Some ideas of these theories are, at least indirectly analyzed in mentioned in the text – e.g. analysis why ineffective (extractive) institutions prevail mention that an institutional change is costly and this cost includes also transaction cost.
Chapter 3.1.2 now clearly specify (in the second paragraph) that one problem connected with the agency theory is that a person can act both as a principal and an agent. There are also other changes both in chapter 3.1 and 3.2 that, as I hope, better analyzes the shortfalls of the agency theory and principal agent theory. Section 3.2.2 concerning the “greasing the wheels” hypothesis shows that corruption can be used as the second best solution of overregulation but it does not remove these obstacles, it only allows individual subjects and no to all society (i.e. to each person) to overcome them and it creates incentives for officials to keep the existing regulations and requirements and to create other. The intention of politicians, officials, or other subjects (including firms benefiting from the barriers) to create and maintain barriers to entrepreneurial activities is an important context of corrupt behavior. Chapter 4 tries to better explain what institutional economics and redistribution system theory can add to both theories. Figure 1 in chapter 4.1.1 is better explained, it is emphasized that the exact value of points A, B and C depends on the specific country’s situation, i.e., there is no universal (world) value for all countries. The value can be affected by moral aware-ness or ethical education of residents of a country, geographical factors, how often con-tacts and relationships between residents occur, how well the organizations fighting against corruption (e.g., police court, anti-corruption agencies) work, etc.
The author.
Reviewer 5 Report
This is an interesting review of the corruption literature and its connection with institutions and institution designs. The review can benefit from adding results from ongoing research on anti-corruption policies using lab and field experiments. There are multiple review papers on that. Additionally, the author should situate some of the discussions to the recent literature. For example, the author mentions situations which have come to be identified as "harassment bribery" in the literature. Related is the use of symmetric and asymmetric punishment mechanisms that should be mentioned in terms of its efficacy. Finally, there is this new work on the Republic of Beliefs which the author should cite especially in their discussion on beliefs and how they affect the corruption environment.
Author Response
Thanks for your valuable comments.
I mention in the text some laboratory experiments. The ideas of “Republic of beliefs” are incorporated in chapter 3.1.2 (page 7 and 8).
The author.
Round 2
Reviewer 2 Report
I believe the paper has significantly mollified my original concerns. It is a bit verbose at times, which could be improved. Also, thorough spell-checking should still be done.
Reviewer 3 Report
The authors have made several changes to the document. I believe these changes answer to my previous questions and improve the quality of the paper. I recommend that the paper is accepted in its current form.
Reviewer 4 Report
Thank you!
The author mention that the main contribution lies in the complex evaluation of standard and alternative approaches to corruption, in emphasizing of importance institutional economics and redistribution system theory for the fight against corruption. By retrospective analyzing the background of the paper, it is not clear how the author fits this objective. The work does not seem to fit the requirements of a reputable journal and requires a more solid methodological background. A principal-agent model of corruption implies a methodological approach, and this can be seen by analyzing the following paper: Groenendijk, N. (1997). A principal-agent model of corruption. Crime, Law and Social Change, 27(3), 207-229.